# Structured Illumination Microscopy Improves Spot Detection Performance in Spatial Transcriptomics

**DOI:** 10.3390/cells12091310

**Published:** 2023-05-04

**Authors:** Alejandro Linares, Carlo Brighi, Sergio Espinola, Francesco Bacchi, Álvaro H. Crevenna

**Affiliations:** 1Epigenetics and Neurobiology Unit, European Molecular Biology Laboratory, 00015 Rome, Italy; 2CrestOptics S.p.A., 00165 Rome, Italy

**Keywords:** structured illumination, spatial transcriptomics, super-resolution, gene expression, in situ sequencing, deconvolution microscopy

## Abstract

Spatial biology is a rapidly growing research field that focuses on the transcriptomic or proteomic profiling of single cells within tissues with preserved spatial information. Imaging-based spatial transcriptomics uses epifluorescence microscopy, which has shown remarkable results for the identification of multiple targets in situ. Nonetheless, the number of genes that can be reliably visualized is limited by the diffraction of light. Here, we investigate the effect of structured illumination (SIM), a super-resolution microscopy approach, on the performance of single-gene transcript detection in spatial transcriptomics experiments. We performed direct mRNA-targeted hybridization in situ sequencing for multiple genes in mouse coronal brain tissue sections. We evaluated spot detection performance in widefield and confocal images versus those with SIM in combination with 20×, 25× and 60× objectives. In general, SIM increases the detection efficiency of gene transcript spots compared to widefield and confocal modes. For each case, the specific fold increase in localizations is dependent on gene transcript density and the numerical aperture of the objective used, which has been shown to play an important role, especially for densely clustered spots. Taken together, our results suggest that SIM has the capacity to improve spot detection and overall data quality in spatial transcriptomics.

## 1. Introduction

Spatial transcriptomics encompasses a recent series of methods that aim to provide molecular maps of the RNA transcriptome of single cells within their natural tissue context [1,2,3,4,5]. Whereas some methods use patterned substrates to capture RNA sequences [6] or fluidics to barcode spatial information within the tissue [7], others consist of light microscopy-based techniques/approaches to visualize the transcripts directly [8,9,10]. A key aspect for all these light-microscopy-based methods is the need to reliably and uniquely identify each individual transcript. In this regard, a common challenge arises when the observed field of view (FOV) becomes saturated with fluorescent targets. This effect, also known as optical crowding, diminishes the ability to effectively visualize and quantitatively identify individual transcripts. Observing a small set of genes becomes technically challenging when the expression level of a few of them or even a single one is relatively high.

Several approaches address this problem in different ways. First, most methods use different wavelengths to partition the set of genes into smaller groups (commonly three or four ‘colors’), which can then be visualized within a single imaging session, e.g., [11]. Methods such as MERFISH [9,12] or seqFISH+ [10] additionally rely on highly sparse encoding, wherein the individual transcript generates a signal in a few of the several performed imaging cycles. While this allows for the identification of a larger set of genes, it also requires a large number of cycles. The seqFISH+ method requires, for example, 80 cycles [10], thereby prolonging the acquisition times and increasing the potential risk of photobleaching. A further problem with performing a large number of cycles is that tissue integrity may degrade over time. Moreover, technical problems related to these relatively complex protocols, such as bubble formation in fluidics or tissue detachment, may result in unsuccessful experiments. On the other hand, in situ sequencing (ISS) methods do not rely on sparsity and instead encode the complete set of genes of interest in a combinatorial manner, which arises from the product of the number of unique wavelengths available and the total number of imaging cycles performed [13]. However, this approach is limited to up to about 200 genes, considering the use of four different color channels and the performing of at least four to six cycles. Beyond wavelengths, another limiting factor in ISS arises from expression level; gene targets are required to have a low- to mid-expression level so that optical crowding is not a major problem.

The physical size of the individual transcripts to be visualized and the resolution of the imaging system used both also contribute to the aforementioned difficulties. Whereas MERFISH or seqFISH+ generate diffraction-limited spots, those generated by ISS are about 1 µm in diameter [8]. This is because, unlike single-molecule methods, ISS involves an amplification step that increases both the size of the spots and the number of fluorescent labelling sites. To achieve this, barcoded padlock probes (PLPs) recognize and bind genetic targets in situ and amplify them by rolling circle amplification (RCA), forming long single-stranded DNA repeats known as rolling circle products (RCPs) [8]. This procedure makes it possible to visualize genetic targets even with standard epifluorescence microscopy and low magnification objectives [8,13], in contrast to MERFISH or seqFISH+, which rely heavily on high-numerical aperture (NA) and high-magnification objectives. While lower magnification objectives allow for the acquisition of larger areas of tissue at a faster rate, resolution is also sacrificed.

In this work we explore the effect of structured-illumination microscopy (SIM) in combination with both low- and high-magnification objectives on the localization of individual fluorescent amplicon spots. In particular, we refer to SIM as a technique that surpasses the diffraction limit of light (sometimes referred to as super-resolution SIM) that is gentle enough to allow for the monitoring of live-cell dynamics with standard fluorophores [14,15]. The use of SIM has revealed that chromatin arranges in chains of mesoscale domains in live cells [16]. Another example of the use of SIM has been to elucidate the formation, growth and internalization of individual clathrin-coated pits and their relationship to cortical actin in living cells [17]. We hypothesized that an improvement in both contrast and resolution provided by SIM would enhance spot-localization performance in ISS experiments while maintaining relatively large FOVs. Overall, we found that SIM increases both spot detection accuracy and efficiency in comparison to diffraction-limited widefield (with or without deconvolution post-processing) and confocal images. The observed improvement is consistent with low-NA 20×, mid-NA 25× and high-NA 60× objectives, respectively. Altogether, the results presented suggest that SIM delivers more flexibility to the choice of imaging modality and magnification level, which allows for the study of samples and genes of diverse density and complexity.

## 2. Materials and Methods

### 2.1. Animals

All experimental subjects were adult male C57BL/6 mice obtained from local EMBL or EMMA colonies or from Charles River Laboratories (Calco, Italy).

### 2.2. Histology

Mice were transcardially perfused with PBS followed by 4% paraformaldehyde in PB. Brains were left to postfix overnight in 4% PFA at 4 °C. Brains were either sectioned in PBS with a vibratome (Leica VT1000s) or cryo-sectioned. For cryo-sectioning, brains were first briefly rinsed in PBS post-fixation, then left in 30% sucrose in PBS for 2 days before flash freezing in pre-chilled isopentane. Frozen brains were sectioned on a cryostat (Leica CM3050s). Coronal sections of 5 µm were taken from areas of interest.

### 2.3. In Situ Sequencing

ISS experiments were performed either with CARTANA (Now 10× Genomics) kit for hybridization-based targeting direct RNA in situ sequencing (HybRISS) or following the hybridization-based in situ sequencing (HybISS) protocol [13,18] using individual components. The difference between these protocols lies within the target molecule for fluorescent labeling, either RNA (HybRISS) or cDNA (HybISS). In the latter case, cDNA needs to be first generated from an mRNA template by reverse transcription. In both methods, however, RNA transcripts are targeted, not genomic DNA. Briefly, both methods rely on the design of oligonucleotide probes (known as padlock probes, PLP), which target specific regions of gene transcripts in a region of interest within the sample. In the PLPs, the binding sequence is split into two halves. The circularized PLP + transcript complex is then ligated and amplified using a DNA polymerase by a process known as rolling-circle amplification (RCA). After amplification, “bridge” oligos containing an ID region specific to each gene and fluorescently labeled probes specific to these bridge oligos are both added for detection. Following instructions from [19], we generated 5 PLPs for the *Actb*, *Gapdh*, *Atp1a3* and *Slc17a7* genes. For the CARTANA kit, samples were processed following manufacturer instructions. Targeted genes in this work are specified in Appendix A. Several neuronal panels from CARTANA were used, in particular the general central nervous system and the inhibitory, excitatory and glia panels.

### 2.4. Imaging

Imaging was carried out using a Nikon Eclipse Ti2 inverted microscope (Nikon Europe B.V., Amsterdam, The Netherlands) equipped with the CrestOptics X-Light V3 spinning disk system, the DeepSIM X-Light super-resolution add-on module (CrestOptics, Rome, Italy), the Celesta multi-mode multi-line laser source (Lumencor, Beaverton, OR, USA), and the Kinetix sCMOS camera (Teledyne Photometrics, Tucson, AZ, USA). The whole imaging set-up was controlled by NIS-Elements Microscope Imaging software version 5.41.00 (Nikon Europe B.V., Amsterdam, The Netherlands). Images were acquired in various channels according to the experiment, using laser excitations at 405, 477, 546, 638 and 749 nm. Appropriate filters for DAPI, GFP, Cy3, Cy5 and Cy7 were used. Stack images were acquired with 0.8 NA CFI Plan Apochromat Lambda D 20× air objective (Nikon Europe B.V., Amsterdam, The Netherlands), 1.05 NA CFI Plan Apo Lambda S 25× silicone oil objective (Nikon Europe B.V., Amsterdam, The Netherlands) or 1.42 NA CFI Plan Apochromat Lambda D 60× oil objective (Nikon Europe B.V., Amsterdam, The Netherlands) with z-steps of 1 µm, 0.5 µm and 0.3 µm, respectively. Widefield spinning disk confocal and DeepSIM super-resolution acquisitions were performed on 1024 × 1024 pixels tile size to cover the same sample area with the three imaging modalities. SIM super-resolution raw images were reconstructed with 25 iterations of the CrestOptics DeepSIM reconstruction algorithm through the NIS-Elements software. For SIM and confocal imaging modes, maximum projections were generated using NIS-Elements software. For widefield, imaging data was first deconvolved as described below.

### 2.5. Deconvolution

Deconvolution was conducted using DeconWolf [20], an open-source, high performance deconvolution software. Point spread functions were generated for each wavelength and each of the numerical apertures of the objective used within DeconWolf following software instructions. Deconvolution ran for a total of 20 iterations with a tile size of 2048 pixels. After deconvolution, a maximum projection was generated using the same package.

### 2.6. Data Analysis

Gene transcript spot detection was performed using the RS-FISH [21] plugin for ImageJ. For all datasets, the optimal parameters were first determined over a subset of images using the interactive mode, with an anisotropy coefficient of 1 and the RANSAC method for spot fitting (see Appendix A). Once determined, these parameters (provided in Appendix B, Table A1) were introduced as a macro in ImageJ for the automatic processing of all images. For every case, the spot intensity threshold was set to 0 and no background subtraction was performed. Once generated, detected spot localizations and intensities were saved as CSV files. Graphs were made using custom code in RStudio (version 2022.12.0 + 353) [22].

## 3. Results

### 3.1. Structured-Illumination Microscopy Enhances Contrast and Improves Resolution of Amplified Individual Transcripts

First, we carried out hybridization-based in situ sequencing (HybISS) [13] on a mouse brain coronal section, targeting four highly expressed genes (*Actb*, *Gapdh*, *Atp1a3* and *Slc17a7*). Only two channels were used for the visualization of all four genes in order to increase the density of distinguishable spots per channel (Figure 1). First, we visualized the sample using widefield (Figure 1, left panels) as a standard method of reference and then using SIM for the same FOVs (Figure 1, right panels). In both cases, a 25× silicon oil immersion plan apochromatic objective with an NA of 1.05 was used. This objective was used mainly because it combines both a relatively high NA and FOV (~7.09 × 10^4^ µm^2^), especially when compared to 20× objectives commonly used for these applications [13]. A detailed examination of the obtained images revealed that the diffraction-limited spots imaged in widefield mode contain significant blur and poor contrast and are not clearly resolved as individual objects (Figure 1, left panels). Using SIM to image the same amplicons yielded individual spots with enhanced contrast and resolution (Figure 1, right panels).

Next, the improvement in image quality provided by SIM was compared to the one achieved when applying deconvolution to the widefield images, as it has been shown to improve spot detection during the image processing of ISS experiments [19]. Although deconvolution on widefield acquisitions increases image contrast by means of a deblurring effect (Figure 1, middle panels), the improvement in resolution is limited compared to that achievable using SIM. Furthermore, a thorough examination of the same FOVs acquired with SIM or with widefield shows that (i) SIM is able to detect a larger number of bright spots compared to widefield (Figure 2A); (ii) most of those bright spots in the SIM acquisitions were spots that had a lower intensity in the widefield image (Figure 2A); and (iii) SIM rescued the spots with low intensity signals that were poorly visible in widefield or, even worse, confused with the background (Figure 2A).

To quantify the potential improvements in spot detection using SIM instead of widefield, images corresponding to thirty different positions of the mouse brain slice were processed using RS-FISH [21]. The quantification of the spots and their relative intensity (Figure 2B) revealed trends similar to those that can be observed from the visual inspection of the images: a higher fraction of bright and dim spots is identified with SIM as a result of the contrast increase achieved with such a structured illumination technique. In this regard, there are two relevant aspects worth noting: (a) a new population of dim spots appeared in the SIM acquisitions compared to the widefield ones (left-most pink bars in Figure 2B), which corresponds to faint signal from previously undetected spots (as depicted by the white arrows in Figure 2A); (b) there was a slight decrease in the intensity of the brightest spots in widefield mode when imaged with SIM (right-most black bars in Figure 2B), which is most likely due to the decrowding of the fluorophores into multiple spots thanks to the resolving ability of SIM over signals in close proximity. Overall, the use of SIM increases the number of detected spots by about 40% (13,756 vs. 9740 per mm^2^ for SIM and widefield respectively, Figure 2B).

Since the resolution improvement provided by SIM may be even more beneficial in areas of higher spot density, we also quantified the number of detected spots with SIM or deconvolution against those detected in widefield for specific FOVs with increasing transcript density (Figure 2C). The use of deconvolution as a pre-processing step in the image analysis pipeline, as demonstrated previously [19], improves spot detection (grey dots in Figure 2C) by about 30%. In contrast, the use of SIM even shows twice as many spots for some FOVs (represented as those measurements that lie in close proximity to the line with a slope m = 2, in Figure 2C), compared to those detected using widefield (pink dots in Figure 2C). It was expected that only an incremental effect of SIM with higher spot densities would be seen; however, even at relatively low densities, the use of SIM increased the number of detected transcripts. The average optical density of imaged transcripts with HybISS was 15,349 ± 3226, 26,750 ± 4780 and 36,059 ± 6357 (mean ± SEM) spots/mm^2^ for widefield, deconvolution, and SIM images, respectively, and for the same FOVs (Figure 2C). These results indicate that the use of SIM in combination with a 25× silicon oil immersion plan apochromatic objective provides an up to two-fold increase in spot detections of amplified single mRNA transcripts.

### 3.2. Spot Detection Performance Is Enhanced for Relatively Low- and Highly Expressed Genes with Moderate Magnification and NA

As the main effect of SIM is more pronounced for higher densities (Figure 2C), we sought to test even denser areas. Therefore, we carried out HybRISS [23], which directly targets the mRNA molecule of the gene of interest instead of its cDNA (HybISS) and, as a consequence, produces more spots per gene [23]. To further increase the observed density, we used a 275 gene panel (Appendix A) encoded in six imaging cycles, so that for any given cycle, there are about 50 genes per channel visualized. For this experiment, we decided to use a 20× objective with 0.8 NA since this is most commonly used for such an experimental design. Here, we again imaged the same FOVs using widefield and SIM consecutively (Figure 3A). Similar to what we observed for the 25×, the use of SIM improved contrast, reduced background and increased resolution (Figure 3B). Upon detailed observation, we noticed that each channel had a particular optical density of spots (Figure 3C). For simplicity, we refer to the grouped genes as *group x*, *group y* and *group z* for the Cy3, Cy5 and Cy7 channels, respectively.

A detailed examination of the 20× images in widefield mode revealed blurry and unresolved spots, which became easier to identify as single entities when imaged by SIM (Figure 3B). As was conducted for the data with the 25×, we used RS-FISH to retrieve the location of the single spots and extract their intensity. We examined in detail the spots detected using the widefield images and compared them against those detected using SIM or deconvolution at the post-processing of the widefield images. It was observed that, similar to the experiment with the 25×, deconvolved images yield more detections than those in widefield mode. However, SIM provides the highest improvement, with a more noticeable effect as spot density increases (Figure 3C, right panels). We chose the detection parameters that could extract the majority (if not all) of the fluorescent objects that could be seen for a variety of images without introducing artefacts or picking up false positives from the background noise (see Methods and Appendix A). Briefly, these parameters (provided in Appendix B) determined the average radius of the true spots that were localized, an intensity threshold to discriminate between the fluorescent objects and the background, and the algebraical constraints that filter out low-quality detections based on radial symmetry and that decrease spot localization errors. Since it would be impractical to determine the best set of values for each image within any dataset separately, a small random subset of images was chosen, and detection parameters were adjusted so that these behaved robustly and accurately for all the selected examples. Once fine-tuned, those parameters were applied to the whole dataset for automatic processing. Based on our data, the use of SIM improves both the resolution and the overall quality of imaged fluorescent objects, which renders more accurate and reliable spot detection (Figure 3C).

While some gene transcripts are distributed more evenly within the cell cytoplasm (yellow and red spots in Figure 3A,B), others localize preferentially to specific regions in tissue or are even confined to subcellular structures such as the nucleus as dense clusters (magenta spots in Figure 3A,B). We were interested in assessing the performance of spot detection in mid-to-low density areas as well as within those clusters (Figure 4A). Therefore, we kept the quantification of the spot numbers and their relative intensity separated between images of spots at lower densities and those at higher-density forming clusters (Figure 4A,B). The average number of detected spots per unit area (10^−3^/μm^2^) were 13 ± 1, 16 ± 1 and 17 ± 1 (mean ± SEM, N = 30) for the lower-density genes, whereas the numbers were 16 ± 1, 34 ± 3 and 52 ± 5 (mean ± SEM, N = 30) for the more densely clustered spots in widefield, deconvolution, and SIM modes, respectively (Figure 4B). A detailed examination of the spot detection as a function of density showed that using SIM allows for the recovery of larger numbers compared to with widefield (Figure 4C). However, at low-to-medium densities, the use of SIM shows no significant difference compared to the use of deconvolution (Figure 4C) at low magnification (in this case, 20×). Spot detection from SIM images recovered almost four times as many spots from the high density and clustered areas compared to that from widefield (Figure 4B,C) and about twice as many when compared to widefield plus deconvolution (Figure 4C). The use of SIM when imaging with a conventional 20× 0.8 NA significantly benefits spot detection at high densities and/or clusters of amplified single RNA transcripts.

### 3.3. High Magnification, High NA and Super Resolution Is Needed for Precise Transcript Localisation in Crowded Regions

Next, we tested whether examination of the sample at a higher magnification would allow for the discovery of more genetic targets within the tissue. Again, we investigated two genes with notable differences in number of transcripts per unit area. For this comparison, both a conventional 20× 0.8 NA air objective and a 60× 1.4 NA oil-immersion objective were used. Several FOVs of the mouse brain tissue processed with the same HybRISS protocol were imaged at 20× and 60× in widefield, spinning disk confocal, and SIM modes, consecutively (Figure 5A and Appendix A). Here, we analyzed the set of genes encoded in the filters for the Cy5 and Cy7 channels (for the gene *groups x* and *y*, respectively). A detailed examination revealed that (i) most *group x* spots at low density are recognized as single entities even with the low-magnification 20× objective and correspond to those observed at the high NA 60× objective, except for a small fraction of *group x* spots that are not distinguishable from the low NA 20× objective (*group x* in Figure 5A,B, white arrows); (ii) the crowded *group y* spots are completely unresolved into a uniform single blob when imaged with the 20× objective in widefield and confocal modes, and they are resolved as individual objects only with either the use of SIM or the high NA 60× objective (*group y* in Figure 5B). To provide another point of comparison, we also imaged the same FOV with spinning disk confocal microscopy (Figure 5A,B, middle panels). As expected, confocality provided an improvement in contrast and out-of-focus light rejection compared to widefield, but with a smaller effect than that provided by SIM. When super-resolution was performed in combination with the 60× objective, many individual spots within these highly crowded areas were then visible (Figure 5B, lower-right panel, *group y* in magenta). While super resolution provides an evident improvement in spot detection performance in low magnification, higher light collection efficiency (i.e., NA) is indispensable for very crowded regions, even in SIM mode.

In Figure 5C,D we show the ratio between the number of localized spots in low magnification SIM mode in comparison to high magnification widefield, spinning disk confocal, and SIM modes for both the low- and high-density genes. In this case, a trend similar to that from the previous experiments performed with the 20× and 25× objectives alone was observed (Figure 5C,D): confocality and SIM increased amplicon localization detection efficiency, with a more notable effect for more densely distributed transcripts (Appendix A). At very low densities (i.e., 1–2 spots per 100 μm^2^), the use of high NA and high magnification, regardless of the imaging modality, has little advantage over the lower NA.

For those genes with a relatively low expression (*group x*), the number of localizations in high resolution mode remains similar to those in low resolution (depicted by the blue line with slope m = 1 in Figure 5C). Beyond this narrow regime, starting at three spots per 100 μm^2^, the use of a higher NA results in a notably higher number of localizations compared to SIM with the 20× (Figure 5D). Widefield and confocal both provide about twice as much localizations, while SIM imaging provides about three times more compared to the 20× images. For denser areas, the effects of a higher NA were even more pronounced (Figure 5D). Widefield provided an improvement in the number of localizations by about 2.7 times (Figure 5D, triangles), while the confocal raised that to about 3.5 times (Figure 5D, empty circles). SIM at 60× imaging with a high NA had the largest effect, with up to 7 times more localizations compared to its 20× counterpart (Figure 5D, filled circles). The commonly used 20× seems to be a poor choice for spatial transcriptomic experiments, as it misses the detection of individual transcripts at all except for very low densities. For any medium-to-high densities, a NA higher than 0.8 is needed to reliably detect spots, as indicated by the white arrows in Figure 5B.

## 4. Discussion

In general, super resolution provided by the SIM module delivers a good enough performance to accurately and reliably detect targeted transcripts in spatial transcriptomics experiments. The performance of SIM is better than that of widefield or confocal imaging methods with or without deconvolution imaging post-processing. The commonly used 20× 0.8 NA objective in combination with super resolution and even spinning disk confocal imaging provides sufficient detail for the identification of, for example, low-expression genes. However, this effect is considerably inferior in comparison to that provided by higher NA objectives. In this regard, the 25× (or 30×) 1.05 NA objective represents a viable alternative that accounts for the needed resolution while maintaining relatively large FOVs and reliable image acquisition times. Moreover, other commercially available options such as the 40× objective series with NAs ranging from 1.25 (silicon oil immersion) to 1.3 and 1.4 (oil immersion) could also provide the ideal resolution needed by imaging-based spatial transcriptomics applications.

For ISS multiplex experiments, transcript identification depends on the correct localization of single fluorescent spots whose position must remain constant throughout all the imaging cycles. In this sense, the increased localization precision delivered by SIM is of great benefit for signal decoding and the identification of individual transcripts. In this work, the average amplicon size was observed to be ~285 ± 206 nm (Appendix A), which suggests that relatively higher densities could be reliably examined in contrast to the previously reported amplicon size [8,23]. In particular, several hundreds of reads per cell could be achieved with the size reported in this work, based on the estimated maximum number of reads for an ISS experiment [8]. Higher densities can arise by looking at a large number of genes or by looking at certain genes with very high expression levels. SIM has, therefore, the potential capacity to improve the dynamic range of gene expression measurements in ISS experiments.

Here, we performed ISS experiments in thin 5 μm tissue sections. Most current commercial instruments offer solutions for thin slices only. In the near future, the field of spatial biology will move towards the transcriptomic profiling of thicker samples and up to whole organs and embryos. These thicker samples will require optical sectioning capacity for volumetric acquisitions. For these needs, spinning disk confocal or light sheet microscopy may be the modality of choice rather than widefield to better reduce the blurring effect of the out-of-focus light along the Z-axis. Although SIM could provide the desired optical sectioning, it may be challenging to image large sample volumes at reasonable times. Nonetheless, SIM can help to untangle high-density areas and resolve numerous spots in close proximity in selected regions. Having the possibility to easily switch between different modalities (from SIM to confocal or from light sheet to SIM) might be helpful in addressing the large variety of samples to be profiled. Recent advances in structured-illumination microscopy within light sheet systems [24] or confocal microscopy [25] could hold the key to the visualization of whole transcriptomes within entire organs and embryos.

## Figures and Tables

**Figure 1 cells-12-01310-f001:**
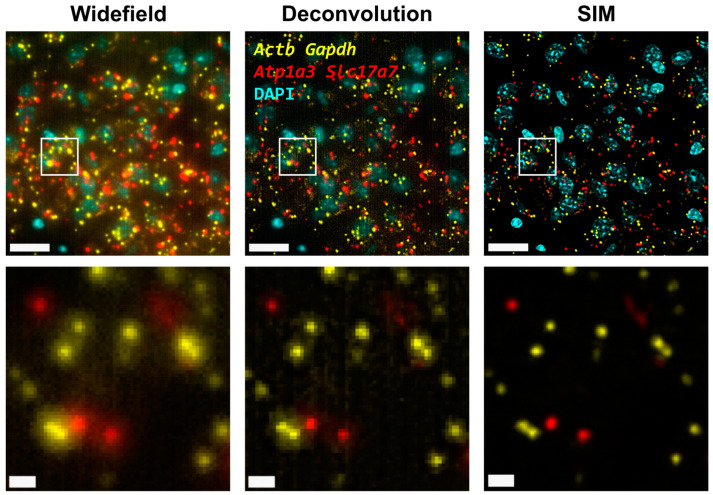
Comparison of gene transcript spot resolution and contrast between widefield (**left**), deconvolution (**middle**) and SIM (**right**). A single FOV of an ISS experiment in a mouse brain coronal section is shown. Sample was stained with DAPI for the nucleus and Cy3 (*Actb* and *Gapdh*) and Cy5 (*Atp1a3* and *Slc17a7*) for gene transcript detection. Lower row corresponds to the white box in the upper row. Scale bars: 20 µm (**upper**) and 2 µm (**lower**).

**Figure 2 cells-12-01310-f002:**
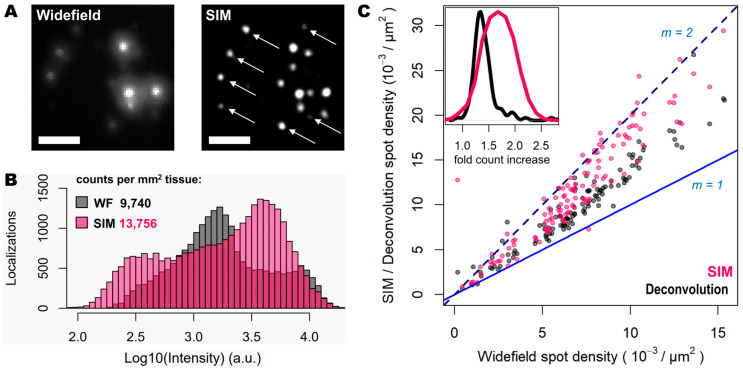
SIM enhances spot detection performance and provides up to twice as many counts. (**A**) Individual transcripts (of *Actb*, *Gapdh*, *Atp1a3* and *Slc17a7*) stained with Cy3 and Cy5 from sample shown in Figure 1, imaged in widefield (left) and SIM (right) modes. White arrows indicate super-resolved spots previously undetected. (**B**) Spot localization count distributions in 2.127 mm^2^ of mouse brain tissue imaged in widefield (grey) and SIM (pink) modes. (**C**) Plot shows the ratio of localization counts for each FOV analyzed (120 in total, 17.72 × 10^3^ µm^2^ each) when comparing widefield against deconvolution (grey dots) and SIM (pink dots). Blue and dotted blue lines correspond to a localization ratio of 1 and 2, respectively. The inner graph shows the distribution of number of fold count increases for deconvolution (black curve) and SIM (pink curve). Scale bars: 5 µm.

**Figure 3 cells-12-01310-f003:**
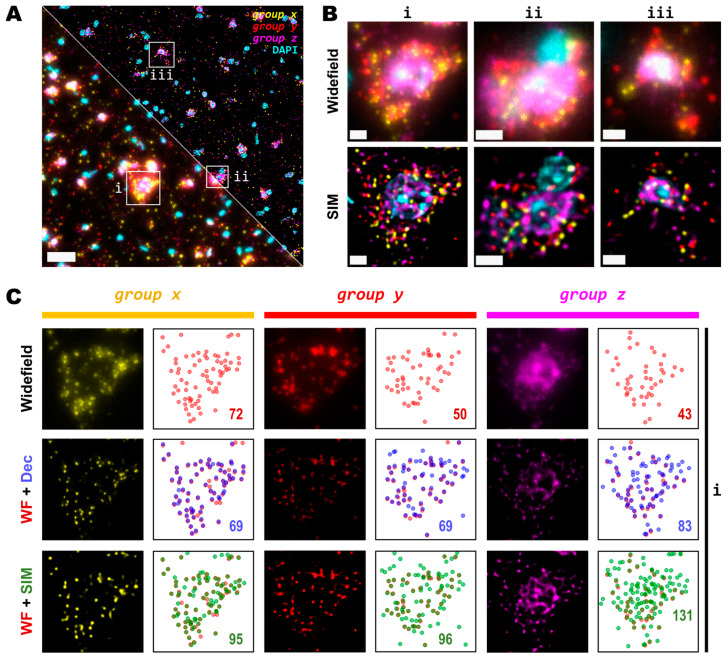
Detection sensitivity of super-resolved spots is increased at moderate magnification and NA. (**A**) Single FOV within a mouse brain coronal section stained with DAPI. Three different sets of 50 genes are visualized in each channel (Cy3, Cy5 and Cy7). Bottom-left and upper-right corners correspond to widefield and SIM imaging modes, respectively. (**B**) Zoomed views of the boxed regions (**i**,**ii**,**iii**) in (**A**) for both widefield (upper row) and SIM (lower row) modes. (**C**) Comparison of resolution and contrast increase for each gene channel from region (**i**) in (**A**) between widefield, deconvolved, and super-resolved images. Next to each fluorescence image of the labelled transcripts, a plot shows the position of all detected spots for widefield (upper row, red), deconvolution (middle row, blue), or SIM (lower row, green) modes. Colored numbers indicate total spot detections in the shown FOV for each channel (*group x*, *group y* and *group z*). Scale bars: (**A**) = 30 µm; (**B**) = 5 µm.

**Figure 4 cells-12-01310-f004:**
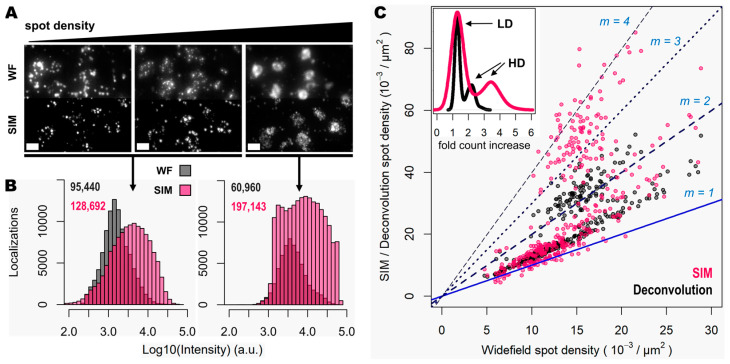
Super resolution enhances detection of highly clustered gene transcripts. (**A**) Three different groups of 50 genes are shown (visualized with the Cy3, Cy5 and Cy7 channels), corresponding to a mouse coronal brain section, imaged in widefield (upper row) and SIM (lower row) modes. A relative increase in gene transcript spot density is shown from left to right. (**B**) Spot localization count distributions in a total of 3.772 mm^2^ of tissue imaged in widefield (grey) and SIM (pink) modes. Left histogram corresponds to the localization count of relatively low-expression genes, imaged in two different channels. Right histogram corresponds to the localization count of a higher-expressed gene that, in contrast, exhibits crowded zones of transcripts that localize preferentially within the cell nucleus (as shown in Appendix A in the case of *gene z*). Total number of localized spots for each condition is shown. (**C**) Plot shows the ratio of spot detection density when comparing widefield against deconvolution (grey dots) and SIM (pink dots) imaging modes. Images corresponding to the genes shown in (**A**) were used for this analysis. In total, 360 images (120 FOVs of 31.43 × 10^3^ µm^2^ for each of the three channels) were analyzed. Blue and dotted blue lines correspond to an increasing spot density ratio from 1 to 4. The inner graph shows the distribution of number of fold count increases for deconvolution (black curve) and SIM (pink curve). The count increase distributions that correspond to low-density (LD) and high-density (HD) gene transcripts are indicated by the black arrows. Scale bars: 10 µm.

**Figure 5 cells-12-01310-f005:**
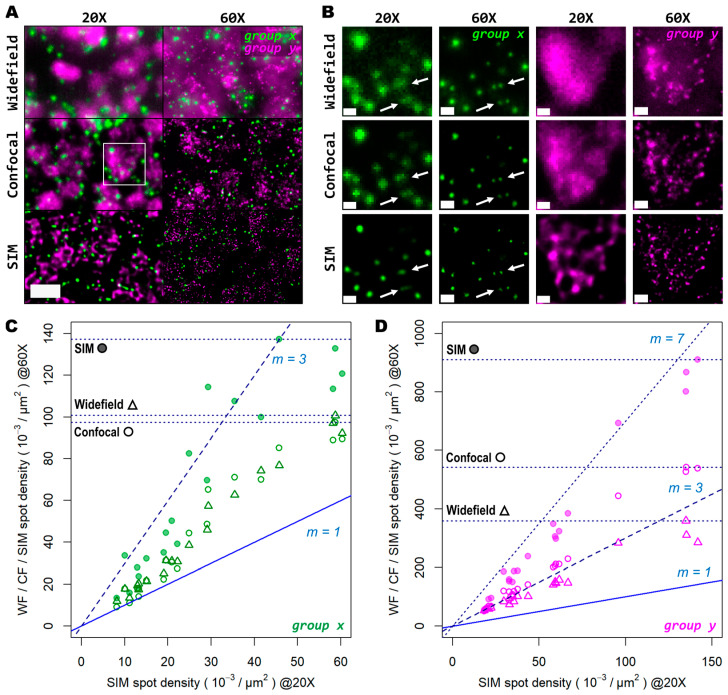
Super resolution in combination with high-magnification high-NA objectives mitigates the effect of optical crowding at various gene transcript densities. (**A**) Two different groups of 50 genes are shown (gene *groups x* and *y*, visualized with the Cy5 and Cy7 channels, respectively), corresponding to a mouse coronal brain section. FOV was imaged with both a 20× (left column) and 60× (right column) objective in widefield (upper row), spinning disk confocal (middle row), and SIM (lower row) modes. (**B**) Gene transcript spot resolution and contrast comparison between the two magnification levels and imaging modes used for both *group x* (left panels, in green) and *group y* (right panels, in magenta). Panels in (**B**) correspond to the region depicted by the white box in (**A**). White arrows indicate a representative example of resolved gene transcript spots provided by increased resolution. Graphs below show the ratio of spot detection density when comparing widefield against confocal (hollow dots) and SIM (colored dots) imaging modes for both the images acquired with the 20× (**C**) and 60× (**D**) objectives. In total, 20 FOVs were analyzed for both magnification levels, each being 32.36 × 10^3^ µm^2^ for the 20× objective (**C**) and 3.54 × 10^3^ µm^2^ for the 60× objective (**D**). Blue and dotted blue lines in (**C**,**D**) correspond to an increasing spot density ratio from 1 to 3. Horizontal dotted lines indicate the maximum spot count gain registered for each imaging mode at 60× compared to SIM at 20×. Scale bars: (**A**) = 10 µm; (**B**) = 2 µm.

## Data Availability

Raw data are available upon request.

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
