# Peer review of "Structured Illumination Microscopy Improves Spot Detection Performance in Spatial Transcriptomics"

_cells, 2023, doi:10.3390/cells12091310_

Round 1

Reviewer 1 Report

This study is very interesting.

Authors can improve by giving a detailed description of methodology used in the paper. 

Give a detailed description of deconvolution of the data. How it is achieved?

Authors specify the exact gene transcripts studied- instead of X, Y and Z. 

Author Response

Q1. Authors can improve by giving a detailed description of methodology used in the paper. Give a detailed description of deconvolution of the data. How is it achieved?

A1. We have improved the description of the methodology used. Changes are now marked in the new version of the manuscript. Additionally, we have added Supplementary Figure S2, which helps explain how the tuning of the spot localization parameter was performed.

Q2. Specify the exact gene transcripts studied, instead of X, Y and Z.

A2. We have now specified the names of the genes for the experiments corresponding to figures 1 and 2. For the experiments corresponding to figures 3, 4 and 5, the name of the 275 genes studied are now specified within the supplementary material. For simplicity, the latter are referred to in the main manuscript as gene groups X, Y and Z.

Reviewer 2 Report

A fluorescent image is a distribution of light-emitting points and is different from a normal image resulting from interference among light waves. Therefore, the resolution of a fluorescent image is an exception of Abbe’s limitation of optical resolution which is derived from interference of light.    Many methods have been developed to improve optical resolution of fluorescent images, and SIM is one of them. 

    SIM microscope becomes commercially available by NIKON since 2015, which provided an ultra-small XY resolution of min 20 nm and Z resolution of 100 nm with the price an order more expensive than a NIKON high-end model of fluorescent microscope. Thus, I never used one unfortunately.

 These authors used the NIKON model to check an improved resolution of model experiments: in citu mRNA hybridization and in situ sequencing (ISS) methods. Compared with the conventional fluorescent image methods unprocessed, the number of detected spots increased respectively by 3-fold and two-fold by SIM and deconvolution that is an older and simpler method to improve resolution. 

              The methods and interpretation of data are sound and have no problems. Presentation of data is excellent and convincing for experimental biologists. 

However, the following weak points make this research an advertise of the commercial model rather than a scientific paper to be published in this journal.

1.    There are many new techniques to improve the resolution of fluorescent, but most of them are not clear on what scientific advance was obtained by the new and usually more expensive method. Although it is nice to get a better photo, but the quality of a photo is not necessarily essential to prove or disprove a scientific hypothesis. An exceptionally successful example is confocal microscopy, which allows us to analyze the position on z-axis, clarifying some hypotheses. Is there any advance for this microscope since 2015? The authors should comment on this. 

In the example of ISS, I would triplicate the measurement rather than using this SIM microscope, because a darker spot may result in due to inefficient round cycling amplification of mis-hybridization, leading to a mis-sequencing. Furthermore, triplication is much cheaper. 

2.    The authors should discuss on the difference between the improvement of resolution by an order of magnitude and the observed increase in the recognizable spots about 3-fold.

    The principle of the improved resolution by SIM is due to the increase of Na with more efficient excitation of fluorophore. A confocal microscope improves resolution by dumping unfocused fluorescent light by a small hole. Thus, it is not suitable for fluorophore rapidly photobleached. In contrast, SIM increases Na by twice and the observed improvement of 3-4-fold seems to be explained just by the increased capture of fluorescent light due to the increased Na. Then, why the increase of optical resolution by an order does not work? 

3.    Because of photobleaching, a fluorophore can emit a limited number of fluorescence photons. The success of IIS greatly depends on the introduction of many photons into a spot by RCA. In contrast, in citu hybridization of transcripts gives the light spots containing various number of fluorophores according to the amount of target RNA. The two examples described here have these different properties. This manuscript is not clear on this difference. Is a single fluorophore observable by this method? Or genomic DNA has been detectedin a similar method? What is the minimum number of fluorophores to be detected? If the authors have this information, it may be great. At least they should discuss on this problem to make this manuscript scientific enough to be published a scientific journal, although this is nice enough to read with the manual of the NIKON model. 

Author Response

Q1. There are many new techniques to improve the resolution of fluorescent images, but most of them are not clear on what scientific advance was obtained by the new and usually more expensive method. Although it is nice to get a better photo, but the quality of a photo is not necessarily essential to prove or disprove a scientific hypothesis. An exceptionally successful example is confocal microscopy, which allows us to analyze the position on z-axis, clarifying some hypotheses. Is there any advance for this microscope since 2015? The authors should comment on this.

A1. There is a complex interrelation, and sometimes not so clear history, between method development and scientific discovery. New methods in general, allow us to observe or measure things that were not possible before. Once we start using them, we begin to explore what new unsolved questions can be answered by them that couldn’t be answered before. Sometimes the effect in a scientific field is immediate (like scRNAseq) and others takes more time (like machine learning or deep learning that had to wait for the development of faster computers and big datasets to be fully realized). Newly developed methods and technology normally have a higher cost than current technology, but as commercial entities scale production and methods mature traditionally the cost of novel technology drops, as exemplified by the history of sequencing of DNA can attest to computers. In particular for SIM, the technology has improved over the past years. One example is that current commercial technology can be faster (lattice SIM of Zeiss) or work for any objective (deepSIM). Of note, since the deepSIM is a module it is much more cost-effective compared to a full standalone system as the ones available from GE, Zeiss or Nikon. Another example is the development of non-linear SIM [Mats G. L. Gustafsson, PNAS, September 2, 2005 102 (37) 13081-13086] and the use of deep learning for image reconstruction [for example: Jin, L., Liu, B., Zhao, F. et al. Deep learning enables structured illumination microscopy with low light levels and enhanced speed. Nat Commun 11, 1934 (2020). https://doi.org/10.1038/s41467-020-15784-x]. Advances like these will impact commercial systems in the years to come.

Q2. In the example of ISS, I would triplicate the measurement rather than using this SIM microscope, because a darker spot may result in due to inefficient round cycling amplification of mis-hybridization, leading to a mis-sequencing. Furthermore, triplication is much cheaper. 

A2. We disagree with the reviewer. Triplicates and the use of SIM have different goals. Triplicates are required to validate an observation. When investigating the spatial transcriptomics of tissues, a triplicate of tissue will be needed to validate the biological interpretation. In our case, a triplicate is the observation of a different field of view. For each measurement we looked at least at 30 different fields of view to make the measurement statistically significant. The use of SIM has the goal to increase the resolution of the image observed. A higher resolution will allow the correct identification of individual fluorescent spots, leading to their correct decoding and therefore a reliable identification of the underlying RNA transcript. In our view, for a biological experiment both, triplicates and SIM are needed. 

Q3. The authors should discuss on the difference between the improvement of resolution by an order of magnitude and the observed increase in the recognizable spots about 3-fold.

A3. We disagree with the reviewer. Linear SIM can only improve resolution by a factor of about 2 [Mats G. L. Gustafsson, PNAS, September 2, 2005 102 (37) 13081-13086]. Non-linear SIM has the theoretical potential to increase the resolution without limit although it has practical limitations given the photon budget required [Mats G. L. Gustafsson, PNAS, September 2, 2005 102 (37) 13081-13086]. We explore the use of linear SIM on in situ sequencing experiments and expect an improvement in resolution not higher than a factor of 2. Note, Nikon does not claim an order of magnitude improvement with SIM (https://www.microscope.healthcare.nikon.com/en_EU/products/super-resolution-microscopes/n-sim-s). A 20nm axial resolution is only possible with single molecule localization, for which Nikon has the N-STORM system, which we do not use, described here: https://www.microscope.healthcare.nikon.com/en_EU/products/super-resolution-microscopes/n-storm-super-resolution.

Q4. The principle of the improved resolution by SIM is due to the increase of Na with more efficient excitation of fluorophore. A confocal microscope improves resolution by dumping unfocused fluorescent light by a small hole. Thus, it is not suitable for fluorophore rapidly photobleached. In contrast, SIM increases Na by twice and the observed improvement of 3-4-fold seems to be explained just by the increased capture of fluorescent light due to the increased Na. Then, why the increase of optical resolution by an order does not work?

A4. We disagree with the reviewer. SIM will only provide an improvement of two-fold for any objective NA used (see above answer). An increased NA of the objective will allow the collection of more light and therefore a higher resolution is achieved. SIM does not have any effect on the objective NA. Of note, a confocal has a minor improvement in resolution, but it has a large effect in contrast [see James B. Pawley, Handbook of Biological Confocal Microscopy]. 

Q5. Because of photobleaching, a fluorophore can emit a limited number of fluorescence photons. The success of ISS greatly depends on the introduction of many photons into a spot by RCA. In contrast, in situ hybridization of transcripts gives the light spots containing various number of fluorophores according to the amount of target RNA. The two examples described here have these different properties. This manuscript is not clear on this difference. Is a single fluorophore observable by this method? Or genomic DNA has been detected in a similar method? What is the minimum number of fluorophores to be detected? If the authors have this information, it may be great. At least they should discuss on this problem to make this manuscript scientific enough to be published a scientific journal, although this is nice enough to read with the manual of the NIKON model.

A5. We disagree with the reviewer. Here, we used only in situ sequencing, in that method several oligo probes (hereafter called ‘padlocks’) are hybridized with RNA transcripts of interest or against cDNA generated from those transcripts, ligated and amplified using rolling circle amplification. The variation we have used was that we used both, experiment for Fig.1 uses oligo probes against cDNA whereas the rest of the paper uses padlocks directly designed against RNA. That change improves the number of dots observed [Lee, Salas, Gyllborg and Nilsson, Sci Rep, 2022]. Once amplified (independently of padlocks targeted against RNA or cDNA), the rolling circle product will recruit many imaging oligos that are fluorescently labeled, generating a fluorescence spot when observed in a microscope. We never targeted genomic DNA. We have tried to make that distinction clearer in the manuscript in page 3, methods, In situ sequencing section. It is not likely that a single fluorophore will be visible using SIM, much less with lower NA of a 20x, as a relatively high number of photons are needed, given the many images required for the image reconstruction process [Mats GL, Gustafsson, DA Agard, JW Sedat, J Microscopy, 1999 Jul;195(Pt 1):10-6; Mats G. L. Gustafsson, PNAS, 2005]. The minimum number of fluorophores needed to be detected will depend on the objective NA. Using the rolling circle amplification several tenths, if not hundreds of fluorophores contribute to the detected signal. That is why single spots are visible with a relatively low NA of a 20X objective, even with a 10X spots are detectable [Lee, Salas, Gyllborg and Nilsson, Sci Rep, 2022 Fig. 2].

Reviewer 3 Report

The authors present the findings of their exploratory research on improving the spot resolution in spatial transcriptomics. The manuscript is well structured and easy to follow. I suggest its acceptance after minor revesions. 

The minor comments are as follows:

1. Introduction lines 75-81 should be a part of result section.

1. Which version of R studio was used for constructing the graphs.

2. Authors are suggested to provide a reference for the same.

3. As the study involves the use of animals, was an ethical approval taken?

4. Results and discussion can be combined as the latter is very short.

Author Response

Q1. Introduction lines 75-81 should be a part of result section.

A1. We disagree with the reviewer. Those lines represent a succinct summary of our findings and are not detailed descriptions of our results.

Q2. Which version of R studio was used for constructing the graphs?

A2. RStudio version (2022.12.0+353) is now specified in the Methods section of the manuscript.

Q3. Authors are suggested to provide a reference for the same.

A3. RStudio is now cited in the manuscript

Q4. As the study involves the use of animals, was an ethical approval taken?

A4. As mentioned in the Institutional Review Board Statement (lines 405-407), for this study an ethical review and approval was not required as mice did not undergo any experimental procedure and were obtained as unwanted production from a different project.

Q5. Results and discussion can be combined as the latter is very short.

A5. The MDPI Cells journal guidelines state that separate Results and Discussion sections are mandatory for publication.

Round 2

Reviewer 2 Report

Q1 is asking what is the new results they got by new technique, namely novelty of this paper.  They could not answer and in the revised manuscript, there is no new claim for the novelty. 

Author Response

We thank the reviewer's and editor's comments over our work.

We have added a paragraph in the Introduction section regarding the advances of SIM microscopy as well as a couple of outstanding examples of its implementation for the study of chromatin and cytoskeletal dynamics, and their relevance in general live-cell imaging.

Worth mentioning is that, in the Discussion section, we talk about the potential gain in gene decoding precision for ISS experiments, due to the increased observed transcript spot resolution delivered by SIM. The mitigation of fluorophore crowding provided by the increased lateral resolution allowed for more precise gene identification in this work. This brings the potential to study genes with a wider spectrum of expression levels within the same tissue/imaging cycle in ISS.

Changes are marked in cyan color in the manuscript.